# Nanomaterial-Based Drug Delivery Systems for Ischemic Stroke

**DOI:** 10.3390/pharmaceutics15122669

**Published:** 2023-11-24

**Authors:** Chengting Jiang, Yang Zhou, Rong Chen, Mengjia Yang, Haimei Zhou, Zhengxiu Tang, Hongling Shi, Dongdong Qin

**Affiliations:** 1Key Laboratory of Traditional Chinese Medicine for Prevention and Treatment of Neuropsychiatric Diseases, Yunnan University of Chinese Medicine, Kunming 650500, China; 18229923764@163.com (C.J.); 15925034542@163.com (M.Y.); 2School of Basic Medical Science, Yunnan University of Chinese Medicine, Kunming 650500, China; chenrongpsyt@126.com (R.C.); 17773572364@163.com (H.Z.); 18087067564@163.com (Z.T.); 3The First School of Clinical Medicine, Yunnan University of Chinese Medicine, Kunming 650500, China; zhou_yang113@163.com; 4Department of Rehabilitation Medicine, The Affiliated Hospital of Yunnan University, Kunming 650021, China

**Keywords:** ischemic stroke, nanomaterials, nanoparticles, drug delivery, targeted therapy

## Abstract

Ischemic stroke is a leading cause of death and disability in the world. At present, reperfusion therapy and neuroprotective therapy, as guidelines for identifying effective and adjuvant treatment methods, are limited by treatment time windows, drug bioavailability, and side effects. Nanomaterial-based drug delivery systems have the characteristics of extending half-life, increasing bioavailability, targeting drug delivery, controllable drug release, and low toxicity, thus being used in the treatment of ischemic stroke to increase the therapeutic effects of drugs. Therefore, this review provides a comprehensive overview of nanomaterial-based drug delivery systems from nanocarriers, targeting ligands and stimulus factors of drug release, aiming to find the best combination of nanomaterial-based drug delivery systems for ischemic stroke. Finally, future research areas on nanomaterial-based drug delivery systems in ischemic stroke and the implications of the current knowledge for the development of novel treatment for ischemic stroke were identified.

## 1. Introduction

Ischemic stroke is a leading cause of death and disability in the world [1]. To date, the effective therapy for ischemic stroke is to restore cerebral blood flow via intravenous thrombolysis or mechanical thrombectomy [2]; however, the therapy is limited to narrow time widows [2], and even timely restoration of cerebral blood flow may not prevent long-term neurological deficits, because of secondary neuronal damage caused by reperfusion [3]. These damaging pathological processes (i.e., inflammation, oxidative stress, and excitotoxicity) persist in the brain during the acute stage and even in the sequelae stage [3]. Thus, targeting these pathological processes has great therapeutic prospects for neuroprotection to improve neuronal survival and outcomes after ischemic stroke. 

At present, numerous neuroprotective drugs have been developed, but almost all failed in clinical translation [4]. There are some important explanations for this translational failure. On the one hand, it lies in the insufficient concentration of drugs that reach the ischemic area because of the limitations of the blood–brain barrier (BBB) [5]. Although a damaged BBB causes an increase in drug penetration during the acute stage of ischemic stroke, it will be closed again long after ischemic stroke [6]. On the other hand, faced with multiple pathological processes, i.e., inflammation, oxidative stress, and excitotoxicity, a single treatment option cannot salvage the ischemic penumbra. Thus, some researchers have developed drug conjugates with synergistic neuroprotection effects for targeting different pathological processes to reduce the damage of neurons to minimize the ischemic area [7,8]. Another important reason is that it is difficult for drugs to target specific damaged cells. There are many different types of cells, i.e., microglia, astrocytes, and neutrophils, involved in pathological processes, and targeting drug delivery to damaged cells may prevent the side effects of drugs [3]. Therefore, novel technologies that can target drug delivery to an intended area or cells, selectively release drugs, and increase the bioavailability of drugs are promising for the treatment of ischemic stroke.

Nanomaterial-based drug delivery systems have gradually been attracting attention in recent years because of their specific characteristics of targeting drug delivery, controllable drug release, biocompatibility, biodegradability, and low toxicity [9]. Recently, many studies have confirmed that nanomaterials can carry drug targets and cross the BBB, after which they can then conduct secondary targeted drug delivery, and then the drug is released to the damaged tissue and cells in ischemic stroke [10]. Therefore, nanomaterials have great prospects in optimizing the treatment of ischemic stroke. There are several reviews that have reviewed drug delivery systems based on thrombus, the brain–blood barrier (BBB), and ischemic brain parenchyma in ischemic stroke, as well as the different approaches for central nervous system (CNS) drug delivery [11,12,13,14]. This review mainly summarized the recent advances in nano-delivery strategies from nanocarriers, targeting molecules and stimulus factors of drug release, aiming to find the best combination of nanomaterial-based drug delivery systems for ischemic stroke. In addition, nano-delivery strategies for future research areas, mainly concerning the generation of oxygen, neuroinflammation, and the diagnosis, as well as novel models of ischemic stroke, will be discussed.

## 2. The Pathogenesis of Ischemic Stroke

Ischemic stroke is a neurological disease characterized by cerebral vascular occlusion. Oxygen depletion, neuroinflammation, and oxidative stress are the main reasons for poor prognoses of ischemic stroke [3,15,16]. Following the interruption of cerebral blood flow, neurons in brain areas dominated by blocked blood vessels will experience hypoxia [15]. Oxygen depletion leads to the irreversible necrosis of neurons, after which a cerebral infarction core is formed [15]. In addition to neuronal necrosis, the BBB is disrupted in ischemic stroke [6]. The BBB is mainly composed of vascular endothelial cells, tight junction proteins (including claudin-5, occludin, and ZO-1), pericytes, and astrocytes [10]. In normal conditions, the BBB could prevent macromolecular substances and peripheral blood cells from entering the brain [10]. Within 120 h of reperfusion after focal cerebral ischemia, BBB permeability is significantly increased, and two peaks appear at 3 h and 72 h [17,18], resulting in leukocytes (i.e., neutrophils and monocytes) in peripheral blood and macromolecular substances entering the ischemic brain, thus further aggravating brain damage [6,19]. The infiltration and accumulation of leukocytes in cerebral ischemic areas depends on the function of adherent proteins expressed on neutrophils, such as integrin *α*M*β*2, CD44, CD11b, macrophage-1 antigen (Mac-1), and lymphocyte function-associated antigen 1 (LFA-1) [20]. During ischemia and reperfusion, intercellular adhesion molecule-1 (ICAM-1), vascular cell adhesion molecule (VCAM)-1, and P-selectin are overexpressed on stressed vascular endothelial cells [6,21]. These molecules can drive leukocytes in peripheral blood to adhere to inflammatory vascular endothelial cells and enter cerebral ischemic areas via membrane-adherent proteins expressed on leukocytes [6,21]. The activation of resident microglia in the brain and the infiltration of leukocytes into the brain result in neuroinflammatory responses that further exacerbate neuronal death [6,22]. In fact, M1-microglia- and leukocyte-mediated neuroinflammatory responses will persist for 1 month after ischemic stroke, which is one of the main reasons for the poor prognoses of ischemic stroke [6,23]. Furthermore, the excessive production of reactive oxygen species (ROS) (i.e., H_2_O_2_, O^2−^, ·OH, and HOCI) after cerebral ischemia onset results in oxidative stress responses, which is also the leading reason for aggravated neuronal death [16,24,25]. Therefore, ensuring the supply of oxygen to the brain and inhibiting neuroinflammatory as well as oxidative stress responses are key to protecting neurons from death and saving ischemic penumbra in ischemic stroke. The abnormally expressed proteins and cells mentioned above may become therapeutic targets for ischemic stroke.

## 3. Nanomaterials Applied in the Treatment of Ischemic Stroke

Reperfusion through intravenous thrombolysis or mechanical thrombectomy is an effective method for treating ischemic stroke [2,26]; however, there is a strict time window limit [2,26]. Even if patients receive reperfusion treatment in time, it is difficult to avoid reperfusion injury after ischemic stroke [3,27]. Reperfusion injury is mainly caused by infiltrated leukocytes and overproduced oxygen radicals after a sudden increase in oxygen, thus resulting in no reflow of blood after the reperfusion of ischemic stroke, which can even worsen neuronal death [27]. Therefore, numerous neuroprotective drugs have been developed to prevent neuronal damage after reperfusion and extend the therapeutic time window to further improve ischemic stroke prognoses [28]. These neuroprotective drugs, such as those with anti-inflammatory and antioxidant stress effects, have been widely studied in animal and cell experiments, and their therapeutic effects have been confirmed in ischemic stroke models [28]; however, these neuroprotective drugs failed in clinical conversion [29,30]. The main reasons for their failure are attributed to an insufficient concentration of drugs in the ischemic brain and the quick elimination of drugs in peripheral blood. Their low absorption, poor bioavailability, poor targeting, and short half-lives result in insufficient concentrations of drugs in cerebral ischemic areas. In view of these limitations, nanomaterial-based drug delivery systems are widely applied in the treatment of ischemic stroke due to their specific properties of targeted drug delivery, controllable drug release, biocompatibility, biodegradability, and low toxicity [10,11,31]. Furthermore, nanoparticles could integrate multiple therapy approaches for ischemic stroke [7,32,33]; for example, Yuan fabricated a nanoparticle with the dual function of antioxidative and anti-inflammatory activities to target the treatment of ischemic stroke [7]. *S. elongatus*, a type of cyanobacteria, is encapsulated in nanoparticles and delivered to cerebral ischemic areas to generate oxygen and absorb carbon dioxide, thus rescuing neurons in the ischemic penumbra area [33]. In addition to delivering drugs to cerebral ischemic areas, leukocyte membrane-derived nanovesicles could inhibit the accumulation of neutrophils and monocytes in ischemic brain areas by consuming the binding sites with neutrophils or monocytes [34]. Therefore, it is essential to understand nanocarriers, the targeting mechanisms of nanoparticles, and drug release mechanisms to further improve the use of nanomaterials in the treatment of ischemic stroke. The possible function and mechanism of nanomaterial-based drug delivery systems in ischemic stroke are showed in Figure 1.

## 4. Properties of Nanomaterials

Nanomaterials are a type of biomaterial with a small size, the same as cellular organelles [31]. Different from traditional macromolecules, nanomaterials have the advantages of a high surface-to-volume ratio, electrical conductivity, magnetization strength, and tunneling effect. Due to these properties, nanomaterials are widely used in diagnosis, therapy, and modeling in the medical field [10,35,36,37]. Based on nanomaterials, drugs can be made into nanoscale particles via nanotechnology, which increases the interaction between drugs and cells [31]. In addition, these drugs can be targeted to a lesion area and controlled for release through designing the surfaces of nanoparticles (Figure 2A), which increases the bioavailability of drugs, reduces drug dosage as well as treatment time, and reduces adverse effects [31]. Furthermore, the tunneling effect of nanomaterials makes it possible for drugs to pass through biological barriers, i.e., the blood–brain barrier and intestinal mucosal barrier, through endothelial cell-mediated endocytosis and paracellular pathways (Figure 2B,C) [10]. These special properties of nanomaterials suggest that they may be a promising therapeutic approach applied in neurodegenerative diseases, i.e., ischemic stroke.

## 5. Nanocarriers in Ischemic Stroke

Nanocarriers serve as carriers to deliver drugs for the treatment of diseases. At present, there are different type of nanocarriers applied for the treatment of ischemic stroke [38,39,40,41,42,43,44,45,46,47,48], including liposomes [38,39], micelles [40,41], poly (lactic-co-glycolic acid) (PLGA) [42,43], dendrimers [44,45], extracellular vesicles (ECs) [46,47], etc. Their advantages, challenges, therapeutic molecules, and injection methods of these nanocarriers in ischemic stroke are showed in Table 1.

### 5.1. Liposomes

Liposomal nanoparticles are submicron capsules with an aqueous core surrounded by lipid bilayers [49]. Lipid bilayers mainly consist of phospholipids and cholesterol, allowing for the encapsulation of lipophilic drugs in the lipid layer, while hydrophilic drugs are encapsulated in the aqueous core [49]. Thus, liposomes provide a multifunctional drug delivery platform that can transport hydrophobic or hydrophilic molecules, including small molecules, proteins, and nucleic acids [49]. In addition, the surfaces of liposomes can be modified with polyethylene glycol and targeting ligands, which enable drugs to target the damaged area [49]. 

At the present, liposomal nanoparticles are widely used in ischemic stroke [8,38,50,51]. Al-Ahmady et al. revealed two windows for therapeutic manipulation based on selective liposomal transport through the blood–brain barrier after ischemic stroke [38]. The research showed that liposomes injected intravenously into mice selectively accumulated in the ischemic brain area at early (0.5 and 4 h) and delayed (24 and 48 h) time points after ischemic stroke [38]. After this, the liposomes are absorbed by microglia 2 to 3 days after ischemic stroke [38]. An interesting discovery showed that pro-inflammatory M1 microglia rapidly increase at 3 days after cerebral ischemic onset, while the M2 microglia that promote brain recovery gradually decrease from this time [52]. In light of these results, liposomal nanoparticles may be used in potential drug delivery to shift the polarization of M1 microglia toward M2 polarization to promote brain repair and block delayed neuroinflammatory responses [38]. However, the research has not confirmed whether liposomal nanoparticles encapsulating therapeutic drugs or molecules could improve outcome after ischemic stroke. Another study makes up for this deficiency [39]; Ahmad et al. fabricated a liposomal drug nanoparticle, which consists of a loaded monosialoganglioside and lipid bilayers [39]. SH-SY5Y and human brain microvascular endothelial cells (HBMECs) were deprived of oxygen and glucose, and MCAO mice were treated with this liposomal drug nanoparticle, and the results showed that it increased the antioxidant protein expressions (HO-1 and NQO1) in both SH-SY5Y and HBMECs, as well as reducing the neurological impairment and infarct volume in mice with ischemic stroke 2 days after the stroke attack [39]. To increase the oral availability of N-butylphthalide (NBP) (a type of water-insoluble drug from celery seeds), Zhang et al. fabricated sodium cholate-appended liposomes to encapsulate the NBP (NBP-loaded CA liposomes) [51]. The research showed that NBP-loaded CA liposomes have the properties of high biocompatibility and controlled drug release [51]. For 12 h after oral administration, NBP-loaded CA liposomes produced higher cumulative release rates (88.09 ± 4.04%) than the NBP group (6.79 ± 0.99%) [51]. In addition, there are an absolute bioavailability (92.65%) and 18.3-fold drug concentration in the brain at 5 min after oral administration in the NBP-loaded CA liposomes group compared with the NBP group in ischemic stroke [51]. Due to these specific properties of NBP-loaded CA liposomes, infarct volume and neurobehavioral deficiency are significantly reduced in the NBP-loaded CA liposomes group compared with the traditional NBP group [51]. Based on the chemotactic characteristics of neutrophils, Tang et al. constructed a nanoparticle consisting of neutrophil membrane-fused nanoliposomal leonurine (Leo), which further increased the accumulation of the drug in the ischemic brain site [53]. Despite the novel therapy of liposomal nanoparticles in ischemic stroke, these studies mainly focus on the treatment of acute ischemic stroke. In fact, neuroinflammation persists in the recovery and sequelae of ischemic stroke. Therefore, it is also important to confirm the effect of liposomal nanoparticles during the recovery period and sequelae of ischemic stroke. Additionally, the gene delivery efficiency of liposomes is much lower than that of viral carriers, which needs to be further overcome [54].

### 5.2. Micelles

The polymeric micelle system was successfully fabricated and applied to improve the physicochemical characteristics of drugs, such as extending the half-lives and reducing the immunogenicity of drugs [8,55]. To overcome the physicochemical characteristics of drugs, the combination of two or more polymers to form micelle nanoparticles is needed [55]. Micelle nanoparticles are formed via the self-assembly of surfactants or amphiphilic polymers in water, and they are widely used as carriers for targeted drug delivery [55]. Song et al. fabricated the micelle nanoparticles by using 1,2-distearoyl-sn-glycero-3-phosphoethanolamine-N-[methoxy (polyethylene glycol)-2000] (DSPE-PEG2000) as the carriers, angiopep-2 (Ang) to modify the surface, and isoliquiritigenin (ISL) to encapsulate in the center of the nanoparticles [40]. Because Ang could bind to the low density lipoprotein receptor-related protein 1 (LRP-1) receptor expressed on the BBB and cross the BBB via LRP-1 receptor-mediated endocytosis [56], the micelle nanoparticles could cross the BBB and accumulate in the ischemic brain areas and then release ISL in the ischemic lesions to alleviate neuronal apoptosis [40]. Furthermore, the half-life of ISL is extended from 0.70 ± 0.22 h to 1.82 ± 0.35 h, and higher bioavailability is confirmed in isoliquiritigenin micelle nanoparticles [40]. In addition to treatment, micelle nanoparticles are also used in disease monitoring and diagnosis [41,57]. To reveal BBB permeability for macromolecules after cerebral ischemia–reperfusion injury, Shiraishi et al. designed a polymeric micelle MRI contrast agent (Gd-micelles) made of poly(ethylene)-block-poly(L-lysine) block copolymers conjugating DOTA-Gd (PEG-P(Lys-DOTA-Gd) [41]. The result showed that Gd-micelles displayed significantly clear contrast images of the ischemic hemisphere at 30 min after intravenous injection compared to traditional contrast agents, which indicated a significant increase in the permeability of the BBB after ischemic stroke [41]. After ischemic stroke, the level of hypochlorous acid (HOCl) will increase in the inflammatory brain area. Based on this fact, Zhang et al. fabricated a nanoprobe comprising three parts: Ang- DSPE-PEG2000 as a carrier to deliver cargo to ischemic brain areas, upconversion nanoparticles as signal reporters, and Cy-HOCl dye as an energy acceptor of UCNPs and the recognition unit of HOCl [57]. The nanoprobe provided brain images when it recognized HOCl in cerebral ischemic areas to monitor neuroinflammation in ischemic stroke [57]. At present, there are few studies about micelle nanoparticle drug delivery systems in ischemic stroke, and more studies are needed to promote clinical conversion.

### 5.3. Poly (Lactic-co-Glycolic Acid) (PLGA)

PLGA, as a synthetic polymer, is widely used as a drug delivery carrier due to its characteristics of biocompatibility, stability, and biodegradability [58,59]. In addition, the safety of PLGA has been confirmed by the United States Food and Drug Administration (FDA) and the European Medicines Agency [58]. At present, PLGA as a nanocarrier has been widely used to deliver drugs or molecules (i.e., RNA) to ischemic brain areas for the treatment of ischemic stroke [37,42,60,61,62,63,64,65]. For example, Choi et al. designed PLGA-based nanoparticles to load and deliver PTEN-induced kinase 1 (PINK1) siRNA (PINK1 NPs) for the treatment of ischemic stroke [42]. PINK1 siRNA NPs controlled the release of PINK1 siRNA and prolonged its elimination time, which manifested as about 87.82% of PINK1 siRNA cumulatively released from the nanoparticles at 72 h [42]. Furthermore, PLGA NPs can be selectively engulfed by microglia, as evidenced by the detection of 61.8% rhodamine+/Iba1+ cells from mice injected with rhodamine-conjugated NPs through fluorescence colocalization imaging, which indicates that the function of microglia may be regulated by PLGA nano-drug delivery systems [42]. The results showed that PINK1 siRNA NPs could decrease infarct volume and behavioral deficiency by inhibiting microglia responses in photothrombotic ischemic mouse models [42]. Despite the above advantage of PLGA, there are still some difficulties that need to be overcome, such as the low drug loading efficiency rate and the difficulty of transforming research results to the clinic [58,60]. Although the above research showed that PLGA nanoparticles are safe, there are still studies indicating that they have moderate toxicity [59,66]. Therefore, more rigorous toxicity experiments need to be designed to further confirm the safety of PLGA.

### 5.4. Dendrimers

Dendrimers are in the fourth class of polymers. Due to their aspects of high transfection efficiency, gene delivery ability, and low cytotoxicity, poly (amide amine) (PAMAM) dendrimers are mainly used as a gene and drug delivery carrier [54,67,68]. Moreover, PAMAM dendrimers also have anti-inflammatory effects [54]. Huang et al. fabricated 1,3-propane sultone (1,3-PS)-modified generation 5 PAMAM (PG5) dendrimers (Au-G5.NHAc-PS) as a carrier, and gastrodin (GAS) was encapsulated in Au-G5.NHAc-PS for the treatment of cerebral ischemia–reperfusion injury [44]. The research showed that Au-G5.NHAc-PS-encapsulated GAS had potential in controlling the release of GAS over a long period of 120 h [44]. Compared to the GAS group, there were better effects in inhibiting inflammatory responses and reducing brain damage in the Au-G5.NHAc-PS/GAS group [44]. Compared with polyamidoamine generation 4 dendrimers (PG4), PG5, and PG6, PG2 as a carrier is more commonly used in treating ischemic stroke, due to its lower cytotoxicity and more efficient drug delivery capability [45,54,68]. For example, Jeon et al. used PAMAM G2-Dexa as a carrier, and fabricated dexamethasone-conjugated PAMAM G2-Dexa for the delivery of the heme oxygenase-1 gene into the cerebral ischemic areas, thus inhibiting inflammation and reducing brain infarct volume [54]. After being injected into hypoxic-ischemic mouse, dendrimers are mostly taken up by microglia, which indicates that dendrimers can serve as a carrier for drug delivery to target microglia [69,70], thus regulating microglia-mediated neuroinflammatory responses. All of the above studies show that dendrimers have a larger size (>100 nm) compared to liposomes and micelles [44,45,54,69,71]. Therefore, further research is needed to confirm whether dendrimers can transport drugs through the BBB in the later stages of ischemic stroke.

### 5.5. Extracellular Vesicles

Extracellular vehicles (EVs) are natural nanoparticles secreted by cells, containing endogenous bioactive molecules, i.e., proteins, RNAs, and DNAs [72,73]. EVs are mainly divided into three types based on their sizes and origins, namely exosomes, microparticles (MPs), and apoptotic bodies [72,73]. As natural intercellular shuttles, compared with traditional nanomaterials, EVs have several advantages, such as biocompatibility, biodegradability, low toxicity, and low immunogenicity [72,73]. However, some unmodified EVs accumulated mainly in peripheral tissues or organs, such as the spleen, liver, and gastrointestinal tract; these different distributions related to the origins of EV cells [73]. Therefore, researchers modified the surfaces of EVs with the rabies virus glycoprotein (RVG), which can target organs rich in acetylcholine receptors, such as the brain [46,73]. Wiklander et al. indicated that EVs significantly accumulated in the brain and heart after the intravenous injection (i.v.) of RVG-EVs into mice [73]. In addition, the research assesses the different routes that injections influence, and found that i.v. showed significantly higher accumulation in the liver and spleen, whereas decreased accumulation was observed in the pancreas and gastrointestinal tract in contrast to intraperitoneal injection (i.p.) and subcutaneous injection (s.c.) [73], indicating that i.p. and s.c may be the optimal choices for digestive system diseases, while intravenous injection is the optimal choice for other systemic diseases. 

Considering these advantages, EVs are considered one of the most promising candidates in nanomedicine in the treatment of ischemic stroke. According to the difference origin, there are several types of EVs applied in the treatment of ischemic stroke, such as neutrophil membrane-derived nanovesicles [37,61,74,75], monocyte/macrophage membrane-derived nanovesicles [32,34], platelet membrane-derived nanovesicles [76], and mesenchymal stem cell-derived exosomes [77].

#### 5.5.1. Neutrophil Membrane-Derived Nanovesicles

Neutrophils play a key role in ischemic stroke and thrombosis. Therefore, neutrophils are important as targets for preventing and treating ischemic stroke [20]. After an attack of ischemic stroke, neutrophils would be recruited and infiltrated into the ischemic brain as early as 12 h, peaking at 1 day, and decreasing until 3 days after stroke onset [19,78]. Based on the chemotactic characteristics of neutrophils, Dong et al. fabricated neutrophil membrane-derived nanovesicles to specifically target the inflamed brain endothelium, and Resolvin D2 (RvD2), which is an anti-inflammatory drug encapsulated in nanovesicles, was delivered to prevent neuroinflammation in mice within 24 h after MCAO surgery [74]. Feng et al. fabricated a nanoparticle-coated neutrophil-like cell membrane from HL-60 cells, and the mesoporous Prussian blue nanozyme is encapsulated in it, and the nanozyme is then delivered and released into inflamed brain microvascular endothelial cells, achieving the function of the reduced recruitment of neutrophils, microglia polarization from M1 to M2, the decreased apoptosis of neurons, and the upregulation of neurogenesis [75]. Based on the interaction between neutrophils and vascular endothelial cells, another study designed neutrophil-camouflaged magnetic nanoprobes (NMNPs), composed of an inner core of superparamagnetic iron oxide (SPIO)-loaded poly (lactic-co-glycolic acid) (PLGA), which served as a highly safe and selective nanoprobe for stroke-induced neuroinflammation imaging [37]. Another interesting study designed a nanoparticle hitchhiking on neutrophils in peripheral blood to deliver drugs to ischemic brain lesions [79]. In particular, the nanoplatform surface is modified by the peptide cinnamyl-F-(D) L-F(CFLFLF) that can specifically bind to the formyl peptide receptor (FPR) located on the surfaces of neutrophils [79]. After intravenous injection on days 2 and 4 after MCAO in mice, the modified nanoparticle can significantly adhere to the surfaces of neutrophils in peripheral blood, and then the nanoparticle was delivered to ischemic brain lesions accompanied by the migration of neutrophils to release ligustrazine for improving ischemic stroke prognosis [79]. Although the neutrophil membrane-derived nanovesicles and neutrophils are good transport carriers in ischemic stroke, studies mainly focus on the acute phase of ischemic stroke, ignoring the recovery phase of ischemic stroke.

#### 5.5.2. Monocyte/Macrophage Membrane-Derived Nanovesicles

Peripheral monocytes quickly infiltrated into the ischemic lesions and differentiated into macrophages after ischemic stroke onset, and the monocytes/ macrophages accumulated into the lesions within 1 day and peak at 3 days [80,81,82]. To suppress the rapid recruitment of monocytes, monocyte cell membrane nanoparticles are designed to directly bind with inflamed vascular endothelial cells, thus reducing the accumulation of monocytes in the peripheral blood in the ischemic brain area [34]. Wang et al. fabricated a monocyte membrane-coated rapamycin nanoparticle (McM/RNP) [34]. After the intravenous injection of McM/RNPs into MCAO/R mice, the accumulation of monocytes on the inflammatory endothelium and microglia proliferation in ischemic brain area were inhibited [34]. In addition, an in vivo biosafety assessment showed that McM/RNPs have no obvious side effects or immunotoxicity after treatment [34]. According to the chemotactic characteristics of monocytes/ macrophages, Li et al. fabricated macrophage membrane-derived nanovesicles with honeycomb manganese dioxide (MnO_2_) on the surfaces of the nanovesicles, and fingolimod (FTY) encapsulated within the nanovesicles [32]. The macrophage membranes of nanovesicles can recognize ICAM-1 and P-selectin overexpressed on the lumen side of damaged vascular endothelial cells to deliver drugs to an ischemic brain [32]. In an ischemic brain, MnO_2_ can consume excess hydrogen peroxide (H_2_O_2_) and convert it into desiderated oxygen (O_2_) to salvage the damaged neurons [32]. Additionally, then, in an acidic lysosome, FTY will be released to promote the transformation of M1 microglia into M2 microglia, eventually inhibiting neuroinflammation and promoting the repair of damaged neurons [32].

#### 5.5.3. Platelet Membrane-Derived Nanovesicles

Platelet (PLT) is the main factor involved in the formation of thrombosis in ischemic stroke, and it is considered a potential target for the treatment of ischemic stroke [47,83,84]. Due to the natural properties of the PLT membrane that could carry specific binding proteins, such as integrins (*α*_6_*β*_1_, *α*_5_*β*_1_, *α*_2_*β*_1_, *α*_v_*β*_3_, and *α*_IIb_*β*_3_), glycoproteins, GPVI, and complementary regulatory proteins (CD59, CD55, CD47, and CD31), platelet membrane-derived nanovesicles could specifically target the damaged blood vessels during the formation of thrombosis [76,83,84]. Inspired by the properties of PLT, Li et al. designed a biomimetic nanocarrier consisting of a PLT membrane envelope loaded with L-arginine and γ-Fe_2_O_3_ magnetic nanoparticles (PAMNs), targeting the thrombus areas to release L-arginine and nitric oxide (NO), thus promoting vasodilation, disrupting the aggregation of PLT, and promoting the recovery of blood flow in obstructed blood vessels in ischemic stroke mice [76]. Wang et al. fabricated nanocarriers comprising a PLT membrane coating inserted with Arg-Gly-Asp (RGD) peptides, which have dual target capabilities that could target the ischemic lesions mediated by the PLT membrane, and target the angiogenic blood vessels mediated by Arg-Gly-Asp (RGD) peptides, thus achieving rapid accumulation in the ischemic brain area [60]. Additionally, human fat extract (FE) encapsulated in the nanocarriers could then be released into the ischemic area to promote angiogenesis and neurogenesis, thus ultimately resulting in an increase in blood flow and the recovery of neurobehavior [60]. There are few studies on platelet membrane-derived nanovesicles in the treatment of ischemic stroke; more studies in vivo and vitro are needed to promote the achievement of bench to bedside in ischemic stroke. In addition, due to the different donors of platelet membrane vesicles [47], further testing and validation about the substances and functions within platelets are needed.

#### 5.5.4. Stem Cell-Derived Nanovesicles

Stem cells (SCs) have been widely used in the treatment of inflammatory diseases because of their capacity to regulate the function of immune cells [77,85]. Exosome, as a novel shuttle, could deliver drugs across the BBB [86]. Therefore, studies fabricated the mesenchymal stem cell-derived exosomes (MSCs) as a nanoparticle carrier for the treatment of ischemic stroke [87,88,89]. Due to MSCs’ poor homing and engraftment into injured tissues, different strategies have been used to modify the surface of MCSs [87,90]. Tian et al. designed the cyclo (Arg-Gly-Asp- D-Tyr-Lys) peptide [c(RGDyK)] on the surface of mesenchymal stem cell-derived exosomes (RGD-exo) to bind to integrin α_v_β_3_ in damaged vascular endothelium cells, thus targeting the ischemic brain [87,91]. Due to overexpressed Chemokine (C-X-C Motif) ligand 12 (CXCL12) in cerebral ischemic lesions, Shi et al. coated C-X-C motif chemokine receptor 4 (CXCR4) onto the surface of MSC membrane vesicles to target cerebral ischemic lesions through the CXCR4–CXCL12 axis [90]. The research also found an interesting phenomenon: the infiltrated peripheral inflammatory cells (i.e., neutrophils and mononuclear macrophages) in the ischemic lesions were cut off and the inflammatory microenvironment improved; the reason for this may be due to the occupation of binding sites of peripheral inflammatory cells by the modified MSC membrane vesicles [90]. Furthermore, A151 (cGAS inhibitor, telomerase repeat sequences) encapsulated in the core of the vesicles could promote the polarization of microglia toward an anti-inflammatory M2-like phenotype through inhibiting the cGAS-STING pathway in microglia [90]. Although the targeting of stem cell-derived EVs has been resolved, the cost associated with their production and validation is enormous for clinical translation, as large-scale stem cell-derived EV isolation requires stem cell replenishment (limited expansion ability) [92,93]. In addition, SC-derived EVs derived from different donors show high heterogeneity, which may influence their features, such as immunophenotypes [94].

### 5.6. Others

There are other polymers used as nanocarriers, such as c-polyglutamic acid (PGA) [95], nanogel [48], chitosan [96], serum albumin [97], small heat shock protein [98], β-cyclodextrin [7], curdlan nanoparticles [99,100], amphiphilic block copolymer (mPEG-b-P(DPA-co-HEMA)-Ce6) [101], gold nanoparticles [35,102,103], etc. For example, Mozafari et al. designed nanogel as a carrier consisting of poly (methacrylic acid) (PMAA) as a carrier and glutathione- (GSH-), located on the surface of PMAA, as a target agent for targeted brain drug delivery [48]. Although nanogel can more easily couple with active targeting agents to increase their targeting compared with other polymers [104], its application in ischemic stroke is still less frequent compared to that of the above nanocarriers.

**Table 1 pharmaceutics-15-02669-t001:** Nanocarriers with their advantages, challenges, therapeutic molecules, and injection methods in ischemic stroke.

No.	Nanocarriers	Advantages	Challenges	Therapeutic Molecules/Cargo	Methods	References
1	Liposomes	High biocompatibility, bioavailability, and no significant cytotoxicity	Poor BBB permeability	N-butylphthalide	Oral administration	[51,105]
2	TPCD nanoparticles (β-cyclodextrin)	High BBB permeability	-	An inflammation-resolving peptide Ac2-26	Intravenous injection	[7]
3	PEG-PLGA	Controlled drug release, good safety, and high biocompatibility and bioavailability	Poor brain-targeting ability	Baicalin	Intranasal administration	[43]
4	Micelles	High bioavailability, biocompatible, relatively non- toxic, and controlled drug release	Poor BBB permeability	Isoliquiritigenin	Intraperitoneal injection	[40]
5	PAMAM dendrimer	High biocompatibility, low cytotoxicity, high gene transfection efficiency, and anti-inflammatory effect	Poor brain-targeting ability	HO-1 plasmid	Stereotaxic injection	[45]
6	PMAA nanogel	High stability, biocompatibility, loading capacity, and controlled drugs release	Poor brain-targeting ability	Edaravone	Intraperitoneal injection	[48]
7	Ca-MOFs	No significant cytotoxicity and immunogenicity, high biocompatibility, and nonviral vectors for miRNA delivery	-	MiR-124	Stereotaxic injection	[106]
8	CeO_2_ NPs	Strong antioxidant capacity, good stability, good compatibility, and controlled drug release	-	Dl-3-n-butylphthalide	Intravenous injection	[103]
9	Honeycomb MnO_2_ nanospheres	Nontoxic, eliminate H_2_O_2_ and generate O_2_, and controlled drug release	Poor brain-targeting ability	Fingolimod	Intravenous injection	[32]
10	Neural progenitor cell-derived EVs	Low immunogenicity, biodegradability, anti-inflammatory effects, and the ability to cross the BBB	Poor brain-targeting ability	-	Intravenous injection	[91]
11	M2 microglial small EVs	The ability to cross the BBB and reduce glial scar formation	-	MiR-124	Intravenous injection	[107]
12	Neutrophil membrane-derived nanovesicles	The ability to cross BBB and inhibiting the recruitment of inflammatory cells	-	Resolvin D2	Intraperitoneal injection	[74]

BBB, blood–brain barrier; TPCD, a cyclodextrin-derived bioactive material; PEG-PLGA, polyethylene glycol–polylactic acid-co-glycolic acid nanoparticles; PAMAM, poly (amidoamine); HO-1, heme oxygenase-1; PMAA, poly (methacrylic acid); Ca-MOFs, Ca-metal−organic frameworks; MiR-124, microRNA-124; CeO_2_ NPs, cerium oxide nanoparticles; MnO_2_, manganese dioxide; H_2_O_2_, hydrogen peroxide; O_2_, oxygen; EVs, extracellular vehicles.

## 6. Target Ligands in Nanomaterials-Based Drug Delivery Systems for Ischemic Stroke

Some of the drug delivery carriers mentioned above lack brain targeting, such as PLGA, micelles, and stem cell-derived nanovesicles [37,40,61,87,88]. Therefore, it is necessary to design target ligands on the surfaces of these nanocarriers to increase the brain targeting of nanoparticles. Based on the different recognition receptors of targeted ligands, these targeted ligands can be mainly divided into three categories, including the ligands targeting damaged vascular endothelial cells, microglia, and neutrophils in peripheral blood.

### 6.1. Targeting of Damaged Vascular Endothelial Cells

Vascular endothelial cell damage is the key cause of BBB disruption and brain edema; while endothelial cells are an important cause of exacerbating brain damage, they also present a new opportunity for the treatment of ischemic stroke due to many targets presented on the damaged endothelial cells. In fact, some proteins are significantly overexpressed on the damaged vascular endothelium in ischemic brain areas, such as integrin *α*_5_*β*_1_ [108], *α*_v_*β*_3_ [88], transferrin receptor (TfR) targeted peptides [95], etc. (Table 2). Based on the overexpressed proteins on the damaged vascular endothelium, many target molecules are designed on the surfaces of nanoparticles for the delivery of drugs to ischemic brain lesions. For example, a synergistic peptide motif, Pro-His-Ser-Arg-Asn (PHSRN), with the ability to support angiogenesis and reduce BBB leakage following ischemic stroke, has the property of binding integrin *α*_5_*β*_1_ [109,110]. Yang et al. fabricated hydroxyethyl starch nanocarriers modified by a synergistic peptide motif, Pro-His-Ser-Arg-Asn (PHSRN), for the targeted delivery of drugs to the ischemic brain area [108]. In addition, stem cell-derived exosomes (SCEs), as carriers for drug transportation, have the advantages of immunomodulatory effects and crossing the BBB, but they also have the property of poor targeting of the brain [87]. Therefore, the cyclo(Arg-Gly-Asp-D-Tyr-Lys) peptide [c(RGDyK)], which has the ability of high affinity to integrin *α*_v_*β*_3_ in damaged vascular endothelial cells, is designed to modify the surfaces of SCEs to deliver drugs to ischemic brain areas [87]. Compared with the above active molecules, the natural neutrophil membranes and platelet membranes have the advantage of biocompatibility, without immunogenicity, long circulation times, and high binding to damaged endothelial cells, serving as better agents with which to modify the surfaces of nanoparticles to transport drugs to the ischemic brain area [37,60,61,83]. There are also some other active molecules that can recognize damaged vascular endothelial cells and be used to modify the surfaces of nanomaterials in ischemic stroke, as shown in Table 2.

### 6.2. Targeting of Microglia

Microglial polarization plays an important role in the progression and prognosis of ischemic stroke [22]. In ischemic stroke, microglia will be rapidly activated and proliferated, and then polarized into two phenotypes: the M1 and M2 phenotypes [22,111]. Among them, M1 microglia have the ability of inducing neuroinflammatory responses, evidenced by the increase in inflammatory factors, such as interleukin (IL)-1β, tumor necrosis factor α (TNFα), and inducible nitric oxide synthase (iNOS), thus aggravating brain damage [22,111]; however, M2 microglia promote brain repair through increasing the release of anti-inflammatory factors and neurotrophic factors, such as IL-4, arginase-1 (Arg-1), and brain-derived neurotrophic factor (BDNF) [22,111]. Therefore, accurately inhibiting the polarization of M1 microglia and promoting the polarization of M2 microglia have important prospects for the treatment of ischemic stroke [22,111]. At present, some studies have fabricated nanomaterial-based drug delivery systems to precisely regulate the function of microglia for the treatment of ischemic stroke [99]. To accurately regulate the function of microglia, some active agents will be used to modify the surfaces of nanocarriers, such as mannose [99,100] and 2-(3-mercaptopropyl) pentanedioic acid (2-MPPA) [69]. For example, mannose, a ligand for recognizing microglia and macrophages, modified the surfaces of curdlan nanoparticles and then accurately delivered nuclear factor-κB (NF-κB) p65 siRNA (sip65) encapsulated in the nanoparticle to microglia in peri-infarct regions [99]. The results revealed a significant transformation from M1 to M2 microglia, and neuroinflammation was inhibited, thus reducing neurobehavior deficit and infarct volume through the silencing of NF-κB p65 in microglia in peri-infarct regions [99]. Moreover, there are reports that liposomes and PLGA nanoparticles can be selectively engulfed by microglia [38,42], which may become a therapeutic strategy for regulating microglia polarization in ischemic stroke.

### 6.3. Targeting of Neutrophils

After ischemic stroke onset, neutrophils would be recruited and infiltrated into the ischemic brain in as little as 12 h [19,78]. Based on the chemotactic characteristics of neutrophils, some active agents on the surfaces of nanoparticles are designed to enhance the adhesion of nanoparticles to neutrophils, and then they hitchhike on neutrophils to reach the ischemic brain areas along with the migration of neutrophils [79,112]. N-acetyl Pro-Gly-Pro (Ac-PGP), an endogenous tripeptide that acts as a ligand with the ability to bind to C-X-C motif chemokine receptor 2 (CXCR2), mainly located on the surfaces of neutrophils, is installed on the surface of dendrigraft poly-L-lysine nanoparticles (DGLNPs) to promote DGLNP adhesion to neutrophils, so that DGLNPs could hitchhike on neutrophils in peripheral blood to deliver drugs to ischemic brain lesions for the treatment of ischemic stroke [112]. In addition to Ac-PGP, cinnamyl-F-(D)L-F(CFLFLF) can also adhere to formyl peptide receptor (FPR) on the surfaces of neutrophils to transport drugs to the ischemic brain area [79], as shown in Table 2.

**Table 2 pharmaceutics-15-02669-t002:** Target molecules with their targets, properties, and carriers in ischemic stroke.

No.	Target Ligands	Targets	Properties	Carriers	References
1	PHSRN peptides	Integrin *α*_5_*β*_1_ enriched in the cerebral vasculature of ischemic tissue	Promoting angiogenesis and reducing BBB leakage	HES	[108]
2	Cyclo (Arg-Gly-Asp-D-Tyr-Lys) peptide	Integrin *α_v_β*_3_ in damaged cerebral vascular endothelial cells	-	Mesenchymal stromal cell-derived exosomes	[87,88]
3	TfR targeted peptides	TfR in cerebral cortex microvessels	A good affinity with the target and smaller in size, without immunogenicity	PGA	[95]
4	PLT membrane and Arg-Gly-Asp peptides	Damaged and angiogenic blood vessels	-	PLGA	[60]
5	Neutrophil membranes	Damaged endothelial cells	Biocompatibility, long circulation times, and without immunogenicity	PLGA	[37,61]
6	Arg-Gly-Asp peptides	Integrin *α_v_β*_3_ in damaged cerebral vascular endothelial cells	-	Neural progenitor cell-derived extracellular vesicles	[91,113]
7	Neutrophil membrane	Inflamed brain microvascular endothelial cells	Biocompatibility, long circulation times, and without immunogenicity	Nanozymes	[75]
8	PLT membrane	Injured vasculature endothelial cells	Biocompatibility, long circulation times, and without immunogenicity	Biomimetic nanobubble	[83]
9	Macrophage membrane	Injured vasculature endothelial cells	Long circulation times, without immunogenicity	MnO_2_ nanosphere	[32]
10	Monocyte membrane	Inflammatory endothelial cells	Inhibiting the recruitment of inflammatory cells to the brain	PLGA	[34]
11	Angiopep-2	Low density lipoprotein receptor-related protein 1 receptor on the BBB	-	Micelles	[40]
12	Mannose	Microglia and macrophages	-	Curdlan nanoparticles	[99,100]
13	2- MPPA	Microglia	-	Dendrimer	[69]
14	The tripeptide agonist N-acetyl Pro-Gly-Pro	CXCR2 receptor on the membrane of neutrophil	Low immunogenicity	DGL nanoparticles	[112]
15	CFLFLF	FPR located on the surfaces of neutrophils	-	PLGA and PEG nanoparticles	[79]
16	Glutathione	The ischemic brain area	-	Nanogel	[48]
17	RVG	Ischemic brain areas	-	EVs	[46]
18	Engineering CXCR4-enriched mesenchymal stem cell membrane vesicles	CXCL 12 in damaged brain	Cutting off the infiltration of neutrophils and macrophage cells in peripheral blood	Polydopamine nanospheres	[90]
19	Sodium cholate	The brain	Enhancing the water solubility of drugs	Liposomes	[51]

PHSRN, Pro-His-Ser-Arg-Asn; BBB, blood–brain barrier; HES, hydroxyethyl starch; Arg-Gly-Asp, arginine-glycine-aspartic; TfR, transferrin receptor; PGA, c-polyglutamic acid; PLT, platelet; PLGA, poly (lactic-co-glycolic acid); MnO_2_, manganese dioxide; 2-MPPA, 2-(3-mercaptopropyl) pentanedioic acid; CXCR, C-X-C motif chemokine receptor; DGL, dendrigraft poly-L-lysine; FPR, formyl peptide receptor; CFLFLF, cinnam-yl-F-(D)L-F; PEG, polyethylene glycol; RVG, rabies virus glycoprotein; EVs, extracellular vehicles; CXCL12, chemokine (C-X-C motif) ligand 12.

## 7. Stimuli-Responsive Nanoparticles in Ischemic Stroke

To reduce the toxicity and side effects of drugs on normal cells, different stimuli-responsive nanoparticles are designed for the accurate release of drugs in damaged brain regions when the nanoparticles reach the ischemic brain area [25,57,101,114,115,116]. These stimuli-responsive nanoparticles mainly include pH-responsive and ROS-responsive nanoparticles.

### 7.1. pH-Responsive Nanoparticles

The pH of ischemic brain areas has been proven to be lower than that of normal brain tissue [117,118,119]. The lower pH in ischemic brain areas may be caused by the rise in lactic acid and CO_2_ in ischemic brain lesions [117,118,119]. Previous studies have shown that the pH of the ischemic lesions decreases to around 6.8 at 4 h after ischemic stroke onset, and the pH fluctuates between 6.5 and 6.9 in the ischemic penumbra [118,120]. To take advantage of the acidic microenvironment in the cerebral ischemic areas, pH-responsive nanoparticles are designed to accurately target drug release, thus reducing the side effects caused by off-target drugs on normal cells [114]. For example, Cheng et al. fabricated pH-responsive rapamycin-loaded nanoparticles for the treatment and diagnosis of ischemic stroke [101]. Magnetic resonance imaging (MRI) and near-infrared fluorescence (NIRF) imaging signals are enhanced in lower pH environments, which enhances the accuracy of drug tracking and ischemic stroke diagnosis [101]. When the nanoparticle is transported to the brain area, rapamycin encapsulated in the nanoparticle would be released in acidic microenvironments in ischemic brain areas, thus inhibiting neuroinflammatory responses and reducing infarct volume [101]. Another study provides a more detailed introduction of pH-sensitive nanomaterial-based drug delivery systems for ischemic stroke [40]: the research fabricated a novel isoliquiritigenin (ISL)-loaded micelle to reduce brain damage via inhibiting cellular autophagy and neuronal apoptosis in ischemic stroke [40]. The release of ISL encapsulated in micelles is pH-dependent, and the release rate of ISL gradually decreases from pH 7.0 to 5.0. Specifically, the release rate of ISL is about 90% at a pH of 5.0 within 48 h, and the lowest ISL release rate (about 60%) is at a pH of 7.0 within 48 h [40]. When nanoparticles are taken up into the lysosomes (pH = 4.5–5.0) of nerve cells, drugs can be rapidly released. pH-responsive nanoparticles prevent the rapid release of drugs into the blood, which results in a prolonged high drug concentration in ischemic areas [40]. In addition to those mentioned above, there are some other drugs encapsulated in nanoparticles released in acidic microenvironments, as shown in Table 3.

### 7.2. ROS-Responsive Nanoparticles

Previous studies have confirmed that reactive oxygen species (ROS) would be excessively released in ischemic brain areas after ischemic stroke or post-stroke reperfusion [25,57,115,116]. These ROS mainly include hydrogen peroxide (H_2_O_2_), hypochlorous acid (HOCl), and hydroxyl radicals (•OH), which caused chronic inflammation and neurodegenerative diseases [25,57,115,116]. Based on the pathological characteristics of the overexpressed ROS in ischemic brain areas, ROS-responsive polymer carriers are designed to target the control of drug release in ischemic stroke [121]; for example, Zhang fabricated an HOCl-activatable upconversion nanoprobe to monitor neuroinflammation in ischemic stroke [57]. After intravenous injection in MCAO mice, the nanoprobe crossed the BBB through transcytosis mediated by low density lipoprotein receptor-related protein (LRP), and it would then be recognized by the overproduced HOCl in ischemic brain areas, thus distinguishing between inflammation and normal brain tissue [57]. Mu et al. designed a nanoparticle modified with reactive oxygen species (ROS)-responsive bond breaking. When the nanoparticle recognized ROS, the nanoparticles would be destroyed and the ligustrazine encapsulated in them will be rapidly released for the treatment of ischemic stroke [79]. Additionally, there are some ROS-responsive nanoparticles, as shown in Table 3.

## 8. Future Research Areas on Nanomaterial-Based Drug Delivery Systems in Ischemic Stroke

Based on the above data, we summarized future research areas on nanomaterial-based drug delivery systems in ischemic stroke into the following five areas. In addition, the current state and prospects of nanomaterial-based drug delivery systems in clinical trials were summarized. 

### 8.1. Generation of Oxygen

Oxygen (O_2_) deficiency is the most fundamental cause of neuronal death in ischemic stroke. Therefore, promoting the generation of O_2_ and consumption of carbon dioxide (CO_2_) in ischemic areas is extremely important for the treatment of ischemic stroke. Based on nanomaterial-based drug delivery systems, there are several studies on the generation of O_2_ through delivering O_2_-generating cyanobacteria or transforming ROS into O_2_ [32,33]. *S. elongatus*, as a type of O_2_-generating cyanobacteria, is delivered to ischemic brain areas and activated to generate O_2_ as well as consume CO_2_ after the irradiation of near-infrared light, thus promoting angiogenesis, reducing neuronal death, behavioral recovery, and reducing brain infarct volume in MCAO mice [33]. Another study fabricated a MnO_2_ nanosphere to convert excess hydrogen peroxide (H_2_O_2_) into O_2_ in ischemic brain areas, thus reducing oxidative stress as well as microglia-mediated neuroinflammatory responses and promoting the survival of damaged neurons in transient MCAO/reperfusion (tMCAO/R) rats [32]. There is currently limited research on the generation of oxygen by nanomaterial-based drug delivery systems for the treatment of ischemic stroke, which requires more research to promote the application of this treatment strategy.

### 8.2. Neuroinflammation

Neuroinflammation plays an important role in the acute and recovery periods of ischemic stroke [122]. The intense neuroinflammatory response mainly resulted in leukocytes rapidly occurring in peripheral blood and microglia within hours after ischemic stroke onset, which further worsened the disruption of the BBB, neuronal death, and neurobehavioral deficiency [122]. In fact, N2 neutrophils, M2 microglia, and M2 monocytes/macrophages are beneficial to promote brain repair and reduce brain infarct volume, while N1 neutrophils, M1 microglia, and M1 monocytes/macrophages will induce neuroinflammatory responses that result in more severe neuronal death in ischemic stroke [78,99,123,124]. Based on these inflammatory pathological mechanisms, nanomaterial-based drug delivery systems could treat neuroinflammatory responses in ischemic stroke from multiple perspectives. Firstly, based on previous studies showing that nanoparticles can hitchhike on neutrophils to be transported to the ischemic brain area [79,112], neutrophils may be regulated to transform into beneficial N2 phenotypes to promote brain recovery through the regulation of encapsulated drugs or genes in nanoparticles, such as fingolimod and miR-124. In addition, hitching on monocytes into ischemic brain areas through designing target ligands to bind to monocytes may be a novel treatment strategy, which needs to be studied further to be confirmed. Secondly, previous studies have confirmed that some nanoparticles entering the brain can be selectively engulfed by microglia [38,42], and some ligand-modified nanoparticles can directly target microglia [69,99,100], which provide a direct treatment strategy for the transformation of microglia from an M1 to an M2 phenotype in ischemic stroke. Thirdly, targeting inflamed vascular endothelial cells not only inhibits inflammatory responses but could also promote angiogenesis through nanoparticles [74,75,108]. Finally, neutrophil membrane-derived nanovesicles could reduce infiltrated neutrophils and monocytes through the consumption of inflammatory binding sites in ischemic stroke [34], which is also a novel strategy for the treatment of neuroinflammation. In addition, previous studies have shown that engineering CXCR4-enriched MSC membrane vesicles could cut off the infiltration of neutrophils and macrophages in peripheral blood to ischemic brain lesions [90]. Therefore, other ligands with the ability of binding to inflamed endothelial cell-modified nanoparticles may also have the ability to cut off the infiltration of leukocytes to ischemic brain areas, which needs to be confirmed.

### 8.3. The Diagnosis of Ischemic Stroke

Magnetic resonance imaging (MRI) is the gold standard for diagnosing ischemic stroke, but it has the limitation that it is difficult to achieve early detection and diagnosis. Accompanied by the application of nanotechnology to ischemic stroke, the earlier diagnosis of ischemic stroke becomes possible [16,36,102]. The previous study used an ROS-responsive near-infrared-II (NIR-II) imaging nanoprobe delineating the ischemic area after ischemic stroke for 30 min, which is earlier than MRI [16]. Additionally, the nanoprobe could also distinguish between ischemic penumbra and core areas [16]. Moreover, a nanoprobe monitoring neuroinflammation was fabricated and used in ischemic stroke [37,57], which may serve as a diagnostic tool with which to guide the application of anti-neuroinflammatory drugs.

### 8.4. A Novel Model of Ischemic Stroke

Ischemic stroke models are important for studying the pathological mechanisms and drug development of ischemic stroke. At present, the stroke models mainly include the suture occlusion model, bilateral common carotid artery ligation model, and photothrombotic (PT) stroke model [8,125,126]; however, these models are not suitable for all research methods, especially considering thrombolytic therapy [35,126]. Based on the nanotechnology, a new ischemic stroke model that uses thrombin to embolize is proposed [35]. This stroke model better imitates clinical stroke models through thrombin to embolize the middle cerebral artery [35]. Specifically, the magnetic nanoparticle (MNP) is flushed into and blocks the middle cerebral artery, caused by the attraction of the magnet located in the common carotid after intravenous injection [35]. Future research about the drug development of ischemic stroke may be more efficient for promoting the conversion of experimental results from preclinical to clinical based on this new model, and more research is needed to further confirm the stability of this model. Additionally, a comprehensive understanding of biodegradability of MNP is also important for the application of this novel model of ischemic stroke. Some studies confirmed that MNP can be degraded in the body [127,128,129,130]. The MNP mainly consisted of iron oxide and various biomaterials. A study explored the dynamic degradation process of the MNP and iron particle [127]. A long half-life of 21 days and the increase in erythrocyte and hemoglobin were observed, but no toxicity in tissues was observed in this study [127]. In addition, shortened half-life may be harmful for the body, because the overloaded Fe^3+^ quickly released from the MNP can result in oxidative stress that can damage tissues and cells [128,129,130]. Therefore, optimizing the surface coating of MNPs to slow down the degradation of the MNP is important to prevent the toxicity induced by the overloaded Fe^3+^. Based on the optimized design of MNPs, thromboembolic models induced by MNPs have important application prospects in ischemic stroke.

### 8.5. Multitargeted Combined Treatment for Ischemic Stroke

Based on nanotechnology, multitargeted combined treatment for ischemic stroke is possible and has great therapeutic prospects. Multiple neuroprotective treatment methods can be integrated to treat ischemic stroke, such as the integration of the generation of oxygen, neuroinflammation suppression, and oxidative stress inhibition [32]. In addition, better combination strategies for treating ischemic stroke may be designed to improve precise drug release, reduce side effects, and maximize the functions of neuronal protection through the design of a combination of nanocarriers, surface ligands, stimulators of drug release, and drugs encapsulated in nanoparticles.

### 8.6. The Current State of Clinical Trials

At present, the efficacy and safety of several biomaterials (i.e., chitosan, detonation nanodiamonds, nano-hydroxyapatite) have been confirmed in clinical trials [131,132,133,134,135]. However, these clinical trials mainly focus on oral diseases and the diseases of osteopathia, cancer, and dermatosis [131,132,133,134,135]. For example, a clinical study showed that a chitosan-based biocompatible dressing completely improved symptoms in 70% of cutaneous leishmaniasis patients after 8 weeks of treatment. In addition, all patients were completely cured after 16 weeks of treatment, and no adverse events were observed [131]. Detonation nanodiamonds (NDs), as nanocarriers, have the characteristics of high scalability and inhibiting bacterial growth [132]. A clinical study confirmed that NDs-embedded gutta percha could prevent reinfection and bone loss in root canal therapy, and no side effects were observed [132]. One clinical study on the application of micromaterials in ischemic stroke are found [136]. The study fabricated lipo-PGI2, a micromaterial-based drug, with lipid microspheres as a carrier and isocarbacyclin methylester (a PGI2 analogue) encapsulated in the microspheres [136]. The results showed that the neurological deficit and mental symptoms were significantly improved in ischemic stroke patients after the intravenous injection of lipo-PGI2, and no side effects were observed [136]. However, we cannot find any previous clinical study about nanomaterial-based drug delivery systems in ischemic stroke. The main reason may be attributed to insufficient preclinical research on the one type of nanoparticle in ischemic stroke. Currently, more studies focus on the exploration of different assembled nanomaterials, but further in-depth preclinical research about the effectiveness, safety, and mechanism on the one type of nanoparticle in ischemic stroke are ignored. On the other hand, heterogeneity of animal models and the sophisticated physiology and structure of the brain may be another reason for the lack of clinical trials in ischemic stroke.

## 9. Conclusions and Perspectives

Nanomaterials can partially overcome the therapeutic limitations of ischemic stroke, such as through extending the half-lives, increasing the bioavailability of, and the targeted delivery of drugs to damaged brain area, thus prolonging the therapeutic time window, reducing the side effects of drugs on normal cells, and improving the treatment efficiency of drugs in ischemic stroke [14]. In this review, we summarized the recent advances in nanomaterial-based drug delivery systems for the treatment of ischemic stroke. Firstly, we summarized the types and characteristics of nanocarriers in the treatment and diagnosis of ischemic stroke, mainly including liposomes, micelles, PLGA, dendrimers, and EVs. Among them, the first four types of nanocarriers have low toxicity and immunogenicity, while EVs, as natural carriers, may become promising drug delivery carriers in ischemic stroke due to their lack of immunogenicity and toxicity (Table 1). In addition, neutrophil membrane-derived nanovesicles can specifically bind to damaged vascular endothelial cells, and, at the same time, they can reduce the infiltration of neutrophils into ischemic brain areas due to the binding sites occupied by nanovesicles [34], which may be more promising nanocarriers for drug delivery in ischemic stroke. Moreover, stem cell-derived nanovesicles (SCs) are more suitable for various immune inflammatory diseases due to their immunoregulation (Table 1). All of the above nanocarriers have the ability to load and transport drugs, but some of them have poor targeting properties, such as PLGA, micelles, and MSC-derived nanovesicles (Table 1). Therefore, we next summarized the targeted molecules located on the surfaces of nanocarriers. These targeted molecules are mainly able to specifically recognize and bind to damaged blood vessels, microglia, and neutrophils in ischemic stroke (Table 2). Among them, there is a therapeutic opportunity related to regulating the transition of neutrophils towards beneficial phenotypes when nanoparticles hitchhike on neutrophils to reach ischemic brain areas along with the migration of neutrophils, thus promoting brain repair. Finally, we divided nanoparticles into two categories based on the factors that stimulate drug release in the ischemic brain area, namely pH- and ROS-responsive nanoparticles.

Based on the studies in the present review, several issues can be selected for investigation. Firstly, most of the studies on ischemic stroke mainly focus on the bioavailability, side effects, and targeting properties of nanoparticles, while there is little research on the toxicity and clearance of nanoparticles in ischemic stroke, which need to be explored further. Secondly, further investigation on the mechanisms and therapeutic effects of nanoparticles in ischemic stroke is required to promote the transformation of nanomaterial-based drug delivery systems from bench to bedside in ischemic stroke. Additionally, there are many studies on different assembled nanoparticles, but there is a lack of in-depth research on the same types of nanoparticles in ischemic stroke. Thirdly, multitherapeutic nanoparticles will be designed for multi-target treatments of ischemic stroke by selecting the optimal combinations of nanocarriers, surface-targeted molecules, and drugs, such as the nanoparticle consisting of neutrophil membrane-derived nanovesicles as a carrier, mannose that can bind to microglia, and macrophages that modified the surface of the nanovesicles [99,100], encapsulating multiple drugs with anti-inflammatory, antioxidant, and anti-apoptotic effects, thus regulating inflamed vascular endothelial cells, promoting angiogenesis, reducing BBB leakage, and regulating the functions and polarization of microglia. This method may provide a new strategy for targeted treatment with traditional Chinese medicine prescriptions for ischemic stroke. Moreover, the chronic neuroinflammatory response that occurs in the later stages of ischemic stroke is an important cause of neurobehavioral deficits in ischemic stroke; however, there is currently a lack of research on nanomaterials in the later stages of ischemic stroke, which will become a promising research area in the future for the treatment of ischemic stroke.

## Figures and Tables

**Figure 1 pharmaceutics-15-02669-f001:**
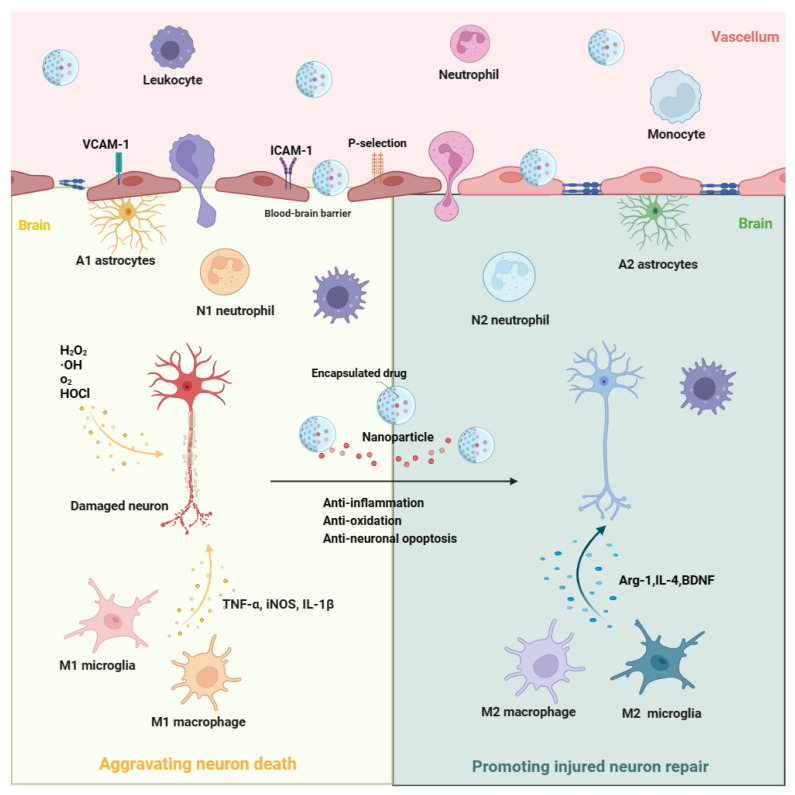
The possible function and mechanism of nanomaterial-based drug delivery systems in ischemic stroke. After ischemic stroke, M1 microglia and A1 astrocyte are rapidly activated in ischemic brain areas. In addition, leukocytes (including neutrophils and monocytes) in peripheral blood are recruited into the ischemic brain area. Among them, M1 microglia, A1 astrocyte, N1 neutrophils, and M1 macrophages would aggravate the neuron death through the overproduction of inflammatory molecules (i.e., TNF-α, iNOS, and IL-1β) and ROS (i.e., H_2_O_2_, O^2−^, ·OH, and HOCI). Based on the nanomaterial-based drug delivery systems, these harmful cell phenotypes may be shifted towards beneficial phenotypes (M2 microglia, A2 astrocyte, N2 neutrophils, and M2 macrophages) to exert the function of anti-inflammation, anti-oxidation, and anti-neuronal apoptosis, thus promoting injured neuron repair in ischemic stroke through the release of drugs or genes that are encapsulated in the nanoparticles. ICAM-1, intercellular adhesion molecule-1; VCAM-1, vascular cell adhesion molecule-1; H_2_O_2_, hydrogen peroxide; •OH, hydroxyl radicals; O_2_, oxygen; HOCl, hypochlorous acid; TNFα, tumor necrosis factor α; iNOS, inducible nitric oxide synthase; IL-1β, interleukin-1β; Arg-1, arginase-1; IL-4, interleukin-4; BDNF, brain-derived neurotrophic factor.

**Figure 2 pharmaceutics-15-02669-f002:**
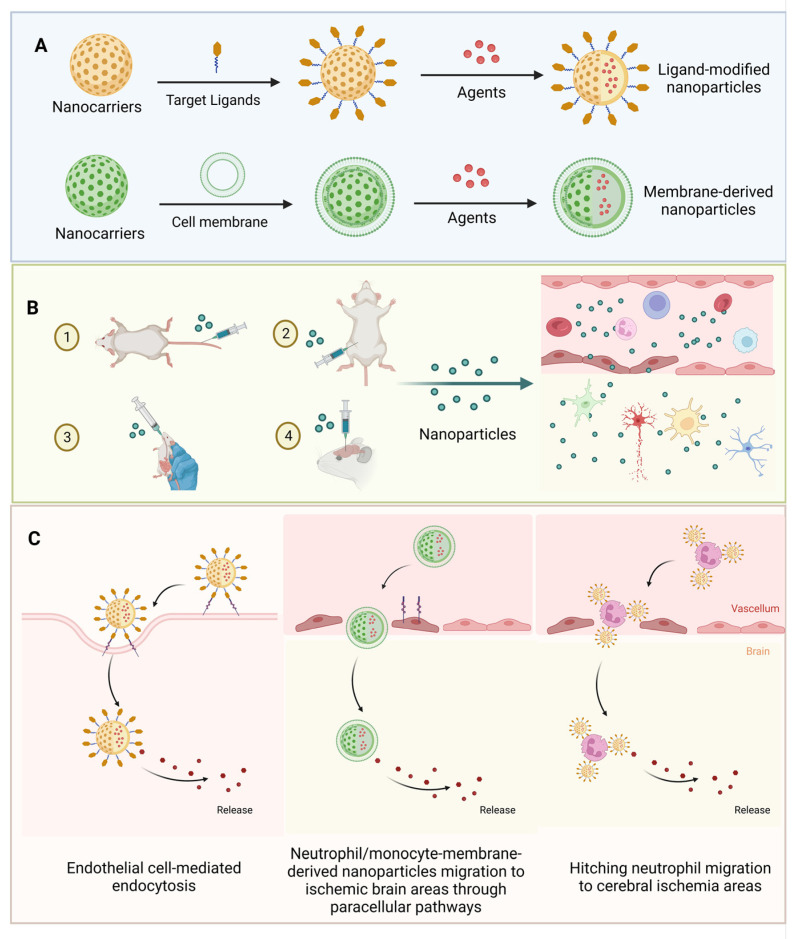
Preparation and delivery of nanoparticles in ischemic stroke. (**A**) The preparation of nanoparticles mainly includes three steps, including the selection of nanocarriers, modification of surface molecules on nanocarriers, and agent encapsulation. Based on the differences in surface molecules, nanoparticles are mainly divided into ligand-modified nanoparticles and membrane-derived nanoparticles. (**B**) Nanoparticles enter the ischemic brain area through different injection methods, mainly including ① intravenous injection, ② intraperitoneal injection, ③ oral administration, and ④ stereotaxic injection. (**C**) Based on the disruption of BBB, the nanoparticles are delivered to the ischemic brain area through endothelial cell-mediated endocytosis and paracellular pathways in ischemic stroke. Among them, the overexpressed receptor located on the damaged BBB can be recognized by ligand-modified nanoparticles, and then the nanoparticles can cross the BBB through endothelial cell-mediated endocytosis. In addition, based on chemotactic characteristics of neutrophils and monocytes, neutrophil/monocytes membrane-derived nanoparticles cross the BBB through paracellular pathways. Moreover, neutrophils in peripheral blood can be bound by ligand-modified nanoparticles, and then the nanoparticles can hitchhike on neutrophils to reach the ischemic brain areas.

**Table 3 pharmaceutics-15-02669-t003:** Stimulating factors for the release of agents encapsulated in nanoparticles in ischemic stroke.

No.	Stimulating Factors for Drug Release	Nanoparticles	Agents	References
1	Acidic environment	Hydroxyethyl starch	Smoothened agonist	[108]
2	Acidic environment	Amphiphilic block copolymer (mPEG-b-P(DPA-co-HEMA)-Ce6)	Rapamycin	[101]
3	Acidic environment	A polyion complex micelle	t-PA and nitroxide antioxidant 4-amino-2,2,6,6-tetramethylpiperidine-1-oxyl	[8]
4	Acidic environment	L-carnosine peptide	Dexamethasone	[63]
5	Acidic environment	Micelle	Isoliquiritigenin	[40]
6	Acidic environment	Ca-Metal−organic frameworks	MiR-124	[106]
7	HOCl	Upconversion nanoparticles	NIR emission	[57]
8	H_2_O_2_	Nanozyme	Fe_3_O_4_	[25]
9	ROS	A pharmacologically active oligosaccharide material nanoparticles	An inflammation-resolving peptide Ac2-26	[7]
10	ROS	A dye-sensitized system between IR-783 dye and lanthanide-doped nanoparticles	NIR-II luminescence imaging	[16]
11	ROS	Polymer PLGA-TK-PEG-peptide	Ligustrazine	[79]

t-PA, tissue plasminogen activator; HOCl, hypochlorous acid; NIR, near-infrared; H_2_O_2_, hydrogen peroxide; Fe_3_O_4_, iron oxide; O_2_, oxygen; ROS, reactive oxygen species; PLGA, poly (lactic-co-glycolic acid); TK, a ROS-cleavable ketal; PEG, polyethylene glycol.

## Data Availability

Data are contained within the article.

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
