# Peer review of "Nanomaterial-Based Drug Delivery Systems for Ischemic Stroke"

_pharmaceutics, 2023, doi:10.3390/pharmaceutics15122669_

Round 1

Reviewer 1 Report

Comments and Suggestions for Authors

The submitted manuscript « Nanomaterial-based drug delivery systems for ischemic stroke” represents a systematic review of the field, with 73 references up to 126 after 2020 and includes a few references for 2023. Despite the huge quantity of data, the authors succeeded to avoid a boring style by introducing informative figures and tables and a detailed “Conclusions and Perspectives” section. The manuscript worth publication in Pharmaceutics

·         Did the authors used an external resource to compose the figures ? If so, please add the information

·         Please define all the abbreviations in the legends of figures when applied (example in figure 1: VCAM-1). Remind that the figures should be readable without referring to the main text

·         Line 539: use CO2 instead of CO2

Author Response

Dear Reviewer 1,

Thank you very much for taking the time to review our manuscript. According to your comments, we have made point-by-point revisions that are marked in red font in the resubmitted revised manuscript. We also upload these responses as a Word file. Please see the attachment.

Comments 1: Did the authors used an external resource to compose the figures? If so, please add the information.

Response 1: The figures are made using the software of Biorender, and the website of Biorender is: https://www.biorender.com/. We have provided this information in Acknowledgement (lines 790-792). In addition, we have obtained the permission for publication from Biorender, and the license certificates of Figure 1 and Figure 2 are provided as follows.

Comments 2: Please define all the abbreviations in the legends of figures when applied (example in figure 1: VCAM-1). Remind that the figures should be readable without referring to the main text.

Response 2: As you suggested, we have defined all the abbreviations in the legends of figures (lines 151-154, lines 457-461, lines 536-541, lines 595-597), that are marked in red.

Comments 3: Line 539: use CO2 instead of CO2.

Response 3: We have made corrections by using CO2 instead of CO2 (line 551, line 607, line 611). In addition, we have corrected all similar errors in the revised manuscript, and all corrections are marked in red.

Yours sincerely,

Dongdong Qin

Yunnan University of Chinese Medicine

Reviewer 2 Report

Comments and Suggestions for Authors

The article is a review of all the nanotechnological studies implemented to take care of the ischemic stroke. In detail the review considers the nanomaterial-based drug delivery systems and analyzes the types of the synthesized nanoparticles as well as the strategies invented by other researchers to treat the thrombus. The study is well designed and organized and the article is well written. There is no criticism toward some studies that use magnetic, non biodegradable nanoparticles that can remain in the body after the drug release triggering nanotoxic effects. It is singular that there is no study on NPS releasing drugs for the distruction of the clot starting from the fibrin dissolution. 

Author Response

Dear Reviewer 2,

Thank you very much for taking the time to review our manuscript. According to your comments, we have made point-by-point revisions that are marked in blue font in the resubmitted manuscript. We also upload these responses as a Word file. Please see the attachment.

Comments 1: There is no criticism toward some studies that use magnetic, non-biodegradable nanoparticles that can remain in the body after the drug release triggering nano-toxic effects.

Response 1: Thanks for your valuable comments. Some studies confirmed that the magnetic nanoparticle (MNP) can be degradable in the body, and a long half-life were observed. In addition, no toxic effects were observed. We have provided this information (lines 676-688).

Comments 2: It is singular that there is no study on NPS releasing drugs for the destruction of the clot starting from the fibrin dissolution.

Response 2: As you concerned, studies on NPS releasing drugs for the destruction of the clot starting from the fibrin dissolution, are still blank. We cannot find any preclinical and clinical studies, which may need further explore in the future research. 

Yours sincerely,

Dongdong Qin

Yunnan University of Chinese Medicine

Reviewer 3 Report

Comments and Suggestions for Authors

The systematic review “Nanomaterial-based drug delivery systems for ischemic stroke” by Jiang et al. is a very interesting manuscript which shows the newest applications of nanoparticles for ischemic strokes.

The authors start with an introduction to the topic and go over to the pathophysiology of stroke. Then the different materials which can be used are described.

Future applications and perspectives are shown in this review.

The review in this specific field is novel and can be published in the journal Pharmaceutics.

However, I still miss the current state in the hospital and the current state of clinical trials for these systems. Specifically for nanoscale systems for stroke treatment.

May you add one paragraph on this topic as well?

Author Response

Dear Reviewer 3,

Thank you very much for taking the time to review this manuscript. According to your comments, we have made point-by-point revisions that are marked in green font in the resubmitted manuscript. We also upload these responses as a Word file. Please see the attachment.

Comments 1: However, I still miss the current state in the hospital and the current state of clinical trials for these systems. Specifically for nanoscale systems for stroke treatment. May you add one paragraph on this topic as well?

Response 1: Thanks for your valuable advice. As you suggested, we have added one paragraph about the current state of nanomaterial-based drug delivery systems in clinical trials, especially for stroke treatment (lines 601-603, lines 698-722).

Lines 601-603: In addition, the current state, and prospects of nanomaterial-based drug delivery systems in clinical trials were summarized.

Lines 698-722: 8.6. The Current State of Clinical Trials

At present, the efficacy and safety of several biomaterials (i.e., chitosan, detonation nanodiamonds, nano-hydroxyapatite) have been confirmed in clinical trials [131-135]. However, these clinical trials mainly focus on the diseases of oral diseases, osteopathia, cancer and dermatosis [131-135]. For example, a clinical study showed that a chitosan-based biocompatible dressing completely improved symptoms in 70% of cutaneous leishmaniasis patients after 8 weeks of treatment. Besides, all patients were completely cured after 16 weeks of treatment, and no adverse events were observed [131]. Detonation nanodiamonds (NDs), as nanocarriers, have the characteristics of high scalability and inhibiting bacterial growth [132]. A clinical study confirmed that NDs-embedded gutta percha could prevent reinfection and bone loss in root canal therapy, and no side effects were observed [132]. One clinical study on the application of micromaterials in ischemic stroke are found [136]. The study fabricated Lipo-PGI2, a micromaterial-based drug, consisted of lipid microspheres as a carrier, and isocarbacyclin methylester (a PGI2 analogue) encapsulated in the microspheres [136]. The results showed that the neurological deficit and mental symptoms were significantly improved in ischemic stroke patients after the intravenous injection of Lipo-PGI2, and no side effects were observed [136]. However, we cannot find any previous clinical study about nanomaterial-based drug delivery systems in ischemic stroke. The main reason may be attributed to insufficient preclinical research on the one type of nanoparticle in ischemic stroke. Currently, more studies focus on the exploration of different assembled nanomaterials, but further in-depth preclinical research about the effectiveness, safety, and mechanism on the one type of nanoparticle in ischemic stroke are ignored. On the other hand, heterogeneity of animal models, sophisticated physiology and structure of brain may be another reason for the lack of clinical trials in ischemic stroke.

Yours sincerely,

Dongdong Qin

Yunnan University of Chinese Medicine
